# DreamMesh4D: Video-to-4D Generation with Sparse-Controlled Gaussian-Mesh Hybrid Representation

**Zhiqi Li**[*,1,2]  **Yiming Chen**[*,1,2]  **Peidong Liu**[†2]

[1] Zhejiang University   [2] Westlake University
{lizhiqi49, chenyiming, liupeidong}@westlake.edu.cn

## Abstract

Recent advancements in 2D/3D generative techniques have facilitated the generation of dynamic 3D objects from monocular videos. Previous methods mainly rely on the implicit neural radiance fields (NeRF) or explicit Gaussian Splatting as the underlying representation, and struggle to achieve satisfactory spatial-temporal consistency and surface appearance. Drawing inspiration from modern 3D animation pipelines, we introduce DreamMesh4D, a novel framework combining mesh representation with geometric skinning technique to generate high-quality 4D object from a monocular video. Instead of utilizing classical texture map for appearance, we bind Gaussian splats to triangle face of mesh for differentiable optimization of both the texture and mesh vertices. In particular, DreamMesh4D begins with a coarse mesh obtained through an image-to-3D generation procedure. Sparse points are then uniformly sampled across the mesh surface, and are used to build a deformation graph to drive the motion of the 3D object for the sake of computational efficiency and providing additional constraint. For each step, transformations of sparse control points are predicted using a deformation network, and the mesh vertices as well as the surface Gaussians are deformed via a novel geometric skinning algorithm. The skinning algorithm is a hybrid approach combining LBS (linear blending skinning) and DQS (dual-quaternion skinning), mitigating drawbacks associated with both approaches. The static surface Gaussians and mesh vertices as well as the dynamic deformation network are learned via reference view photometric loss, score distillation loss as well as other regularization losses in a two-stage manner. Extensive experiments demonstrate superior performance of our method in terms of both rendering quality and spatial-temporal consistency. Furthermore, our method is compatible with modern graphic pipelines, showcasing its potential in the 3D gaming and film industry. The source code is available at our website: https://lizhiqi49.github.io/DreamMesh4D.

## 1  Introduction

The emergence and development of Generative Artificial Intelligence (GenAI) have significantly revolutionized 3D generation techniques in recent years [20]. The technology has effectively allowed the creation of static objects, including their shape, texture, and even an entire scene from a simple text prompt or a single image. Recently, the wave of advancement has been propelled to the filed of dynamic (4D) content generation [45], which offers immense potential in fields including, but not limited to, AR/VR, filming, gaming and animation. However, it's still quite challenging to efficiently

---

[*]Equal Contribution;    [†] Corresponding author.

38th Conference on Neural Information Processing Systems (NeurIPS 2024).

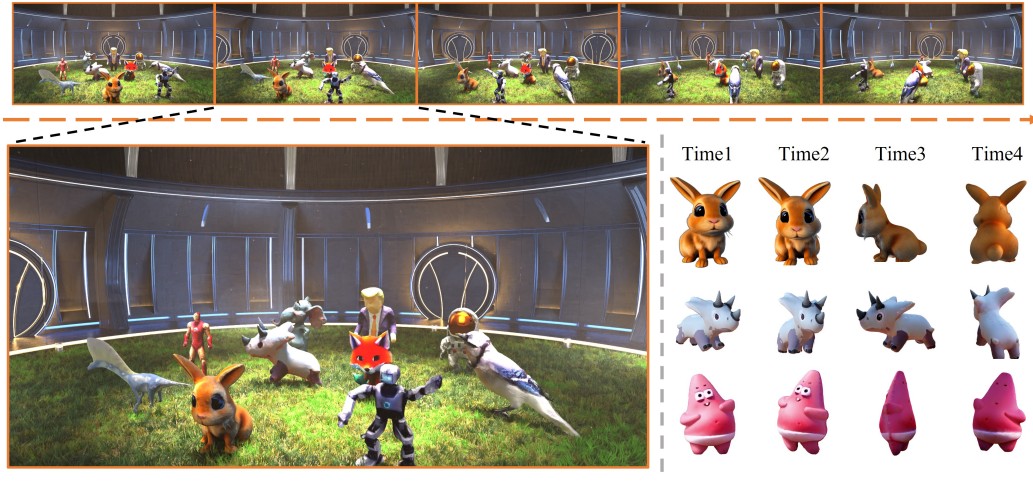

Figure 1: Given monocular videos, our method is able to generate high-fidelity dynamic meshes. We also produce a composited scene demo (top bar and left side of the figure) with the generated dynamic meshes, showcasing our method's compatibility with modern 3D engines.

generate high-quality 4D content due to its increased spatial-temporal complexity and higher demand on algorithm design.

The promising strides in 3D generation are largely attributed to the pre-trained large 2D diffusion models [39, 41, 31]. In particular, score distillation sampling (SDS) [33] enables the 3D generation [30, 55, 18, 42, 9] from scratch by distilling 3D knowledge from a pre-trained 2D diffusion model [39]. Following works on text-to-3D [24, 57, 44, 15] and image-to-3D [26, 35, 50, 48] have further improved the performance of 3D generation tasks both in quality and stability. Inspired by the successes of SDS-based 3D generation, recent works [69, 1, 38, 12, 65, 66] explore generating 4D assets by distilling prior knowledge from pre-trained video diffusion models [54, 3] or novel-view synthesis models [26, 43]. Both text-to-4D [45, 1, 25] and image-to-4D methods [69] mainly rely on pre-trained video diffusion models, which are not yet capable of generating high-quality video, thus usually struggle to generate high-quality 4D content. On the contrary, video-to-4D methods [38, 12, 66, 65] directly generate 4D assets from off-the-shelf monocular videos, making the results more appealing and with better spatial-temporal consistency. Existing video-to-4D methods either rely on the implicit dynamic NeRFs [12] or explicit dynamic Gaussian splatting [38, 65, 66] as the underlying representation. Nevertheless, both of them do not have tight constrains for surface, leading to redundant optimization space and impeding the learning of deformation.

Inspired by modern graphic pipelines for 3D animation, we propose **DreamMesh4D**, which exploits 3D triangular mesh representation and sparse-controlled geometric skinning methods [47, 16] for video-to-4D generation. To better supervise the generation with 2D signals, instead of using classic mesh with UV texture maps, we choose a hybrid representation, SuGaR [9], which marries 3D Gaussians to mesh surface for more elaborate appearance modeling. Flat Gaussians are bound to mesh faces based on barycentric coordinates hence the rendering process of 2D images is differentiable with respect to both the position of mesh vertices and the attributes of Gaussians. For high-quality object modeling and efficient motion driving of the object, our method is designed in a static-to-dynamic optimization manner. In particular, during the static stage, an initial coarse mesh is generated utilizing existing image-to-3D generation methods [26, 68, 61]. Then we refine its both geometry and texture by jointly optimizing the mesh vertices as well as the attributes of bound surface Gaussians under the hybrid representation via both the reference image photometric loss and the SDS loss. For dynamic learning, we uniformly sample sparse control nodes from the surface of the refined mesh, to build a deformation graph. Then at each timestamp, transformation associated with each control node is predicted by a deformation network. The deformation of all mesh vertices and surface Gaussians are obtained from those predicted transformations via a novel geometric skinning algorithm, which benefits from both the LBS (linear blending skinning) and DQS (dual-quaternion skinning) methods. The deformation network is optimized under the supervision of photometric loss from reference video frames, novel-view SDS loss and geometric regularization terms.

Extensive experiments are conducted and demonstrate that our method can generate high-fidelity dynamic textured mesh from monocular video, and significantly outperforms previous works both quantitatively and qualitatively, establishing new benchmark in the field of video-to-4D generation. As shown in Fig. 1, our generated assets can be directly simulated in modern 3D engines, showcasing its potential in the 3D gaming and film industries.

## 2 Related Work

**3D Generation** Since the introduction of score distillation sampling (SDS) by DreamFusion [33], subsequent works [24, 5, 57, 35, 48, 49, 23] have significantly improved the performance of optimization-based 3D generation algorithms. Many works adopt a multi-stage optimization strategy [24, 5, 57, 48] to enhance generated appearance. Another line of research [26, 44, 27, 19] focuses on training multi-view diffusion models to inject multi-view supervision into SDS loss for addressing the Janus problem. DreamGaussian [49] and GaussianDreamer [64] pioneer the usage of 3D Gaussian[18] as the underlying representation and achieve 3D content generation in minutes. Although these methods demonstrate the potential of 3D Gaussian in 3D content generation, obtaining an object with high-quality geometry is still quite challenging. In this work, we explore to employ a Gaussian-mesh hybrid representation [9] in our 4D generation tasks for better modeling of both object surface geometry and texture.

**4D Representations** Dynamic 3D representations form the foundation of 4D reconstruction and generation tasks. Most current methods extend static NeRF [30] to dynamic scenarios. These approaches, such as deformable [32, 34, 51, 59] and time-varying [4, 7, 8, 6] NeRF-based methods, are prevalent. There are also some works trying to model dynamic with 4D neural implicit surface [56, 14]. However, these implicit representations suffer from long optimization time and low reconstruction quality due to its computationally expensive volume rendering and implicit representation. Recent interest in 4D tasks has focused on 3D Gaussian representations due to their fast rendering speed and explicit nature. Some works [63, 58, 22] train networks to predict Gaussian kernel deformations, while others model kernel motion via polynomial representation or per-frame optimization [28, 21]. Besides, both SC-GS [11] and HiFi4G [13] employ sparse control points for Gaussian kernel deformation, with SC-GS using LBS and HiFi4G using DQS to drive Gaussians motion, thus ensuring spatial-temporal consistency. In this work, we propose to deform the object through a novel geometric skinning method, handling the artifacts associated with both LBS and DQS.

**4D Generation** Although great progress has been achieved in 3D generation tasks, 4D generation is still challenging due to its requirement on additional temporal supervision. Since current pre-trained video diffusion models [3, 54] still struggle to generate high-quality video contents, it is challenging to distill useful motion knowledge via SDS optimization. Hence, the performance of existing text-to-4D [45, 1, 25] and image-to-4D methods [69] usually struggle with low appearance quality. Another line of works focus on video-to-4D generation [12, 38, 65]. These methods leverage current multiview diffusion models [26, 43, 27] to calculate the SDS loss [38, 12, 60] or generate per-frame multi-view images [66, 62] as supervision signal. Among them, a concurrent work of our method, SC4D [60], optimizes a set of sparse-controlled dynamic 3D Gaussians by per-frame SDS loss from Zero123 [26] with a coarse-to-fine strategy. However, the issue of unsatisfying surface quality caused by 3D Gaussian-based representation still exists. In contrast, our method is grounding on a Gaussian-mesh hybrid representation [9], enhancing the reconstruction quality both in texture and geometry.

## 3 Method

In this section, we first introduce the relevant preliminaries in section 3.1. Then we illustrate the details of DreamMesh4D which is divided into static stage and dynamic stage in section 3.2 and 3.3 respectively. The overview of our method is presented in Fig. 2.

### 3.1 Preliminaries

**Geometric Skinning Algorithms** Given a mesh with $N_v$ vertices, $\mathcal{M} = \{\mathcal{V}, \mathcal{F}\}$ where $\mathcal{V} = \{\mathbf{v}_i | \mathbf{v}_i \in \mathbb{R}^3\}, i \in \{1, 2, ..., N_v\}$ is the set of vertices and $\mathcal{F}$ represents the triangle faces. In geometric skinning algorithms, there are also some sparse control nodes (bones/skeletons) $\mathcal{P} =$

$\{\mathbf{p}_i | \mathbf{p}_i \in \mathbb{R}^3\}, i \in \{1, 2, ..., N_p\}$, where $N_p$ is the number of control nodes. For a mesh vertex $\mathbf{v} \in \mathcal{V}$, its deformation is calculated by blending a number of neighboring control nodes $\mathcal{N}(\mathbf{v})$ through skinning algorithms. The local deformation for a control node $\mathbf{p} \in \mathcal{N}(\mathbf{v})$ can be decomposed into a deformation matrix $F_{\mathbf{p}} \in \mathbb{R}^{3 \times 3}$ and a translation vector $t_{\mathbf{p}} \in \mathbb{R}^3$, and the deformation matrix can be further decomposed into a rotation matrix $R_{\mathbf{p}} \in \mathbb{R}^{3 \times 3}$ and a shear matrix $S_{\mathbf{p}} \in \mathbb{R}^{3 \times 3}$ using polar decomposition. The strength of the influence of node $\mathbf{p}$ to vertex $\mathbf{v}$ can be represented as $w_{\mathbf{p}}$ and $\sum_{\mathbf{p} \in \mathcal{N}(\mathbf{v})} w_{\mathbf{p}} = 1$. Linear blending skinning (LBS) [47] computes the deformation of vertex $\mathbf{v}$ by linearly blending the influence of nodes in $\mathcal{N}(\mathbf{v})$:

$$\tilde{\mathbf{v}}^{lbs} = \sum_{\mathbf{p} \in \mathcal{N}(\mathbf{v})} w_{\mathbf{p}}(F_{\mathbf{p}}\mathbf{v} + t_{\mathbf{p}}). \tag{1}$$

LBS is widely used due to its simple formulation and natural animations. However, it suffers from the well-known "volume loss" or "candy wrapper" artifacts under complex deformation. As an enhancement, dual-quaternion skinning (DQS) [16] represents the transformation of node $\mathbf{p}$ with a unit dual-quaternion $\zeta_{\mathbf{p}} = DQ(R_{\mathbf{p}}, t_{\mathbf{p}})$. Then the deformation of $\mathbf{v}$ can be calculated with DQS by:

$$\tilde{\mathbf{v}}^{dqs} = \bar{\zeta}\mathbf{v}\bar{\zeta}^*, \text{ where } \bar{\zeta} = \frac{\sum_{\mathbf{p} \in \mathcal{N}(\mathbf{v})} w_{\mathbf{p}}\zeta_{\mathbf{p}}}{||\sum_{\mathbf{p} \in \mathcal{N}(\mathbf{v})} w_{\mathbf{p}}\zeta_{\mathbf{p}}||}, \tag{2}$$

where $\bar{\zeta}^*$ represents the conjugate of $\bar{\zeta}$. DQS can eliminate the artifacts associated with LBS, but cannot handle non-rigid deformation since the strain effect is not considered. It also suffers from an undesirable joint-bulging artifact while blending, which requires artistic manual work to fix [40].

**3D Gaussians and SuGaR**  Gaussian Splatting [18] represents the scene as a collection of 3D Gaussians, where each Gaussian $g$ is characterized by its center $\mu_g \in \mathbb{R}^3$ and covariance $\Sigma_g \in \mathbb{R}^{3 \times 3}$. The covariance $\Sigma_g$ is parameterized by a scaling factor $s_g \in \mathbb{R}^3$ and a rotation matrix represented via a unit quaternion $q_g \in \mathbb{R}^4$. Additionally, each Gaussian maintains opacity $\alpha_g \in \mathbb{R}$ and color features $c_g \in \mathbb{R}^C$ for rendering via splatting. Typically, color features are represented using spherical harmonics to model view-dependent effects. During rendering, the 3D Gaussians are projected onto the 2D image plane as 2D Gaussians, and color values are computed through alpha composition of these 2D Gaussians in front-to-back depth order. While the vanilla Gaussian Splatting representation may not perform well in geometry modeling, SuGaR [9] introduces several regularization terms to enforce flatness and alignment of the 3D Gaussians with the object surface. This facilitates extraction of a mesh from the Gaussians through Poisson reconstruction [17]. Furthermore, SuGaR offers a hybrid representation by binding Gaussians to mesh faces. A SuGaR mesh can be represented as $\mathcal{S} = \{\mathcal{V}, \mathcal{F}, \mathcal{G}\}$ where $\mathcal{G}$ denotes all surface Gaussians. For a triangle face $f \in \mathcal{F}$, the Gaussians $\mathcal{G}(f)$ bound on $f$ are defined with barycentric coordinates. This hybrid representation allows joint optimization of texture and geometry through backpropagation.

## 3.2 Static Stage

The purpose of the static stage is to generate a refined Gaussian splats bounded triangular mesh. This procedure starts with a coarse mesh generated from a reference image $I^*$ that is sampled from all video frames $\mathcal{I}$. Although there exists several fast mesh generation methods [68, 61], we refer to Zero123-based SDS optimization for its stability. In particular, we conduct simple SDS training on a set of randomly initialized 3D Gaussians for a fixed number of steps with regular densification and pruning. After that, we stop densification and pruning, and include SuGaR's regularization terms [9] to make Gaussians aligned to object surface. Finally all Gaussians with opacity lower than a threshold $\bar{\sigma} = 0.5$ are pruned, after which Poisson reconstruction [17] is performed to extract a coarse mesh.

On the generated coarse mesh, we attach $x = 6$ new flat Gaussians to every triangle face. For each training step, we render RGB image $\hat{I}^*$ and mask $\hat{M}^*$ under reference view to calculate RGB loss $\mathcal{L}^s_{ref} = ||\hat{I}^* - I^*||^2_2$ and mask loss $\mathcal{L}^s_{mask} = ||\hat{M}^* - M^*||^2_2$. The SDS loss $\mathcal{L}^s_{SDS}$ based on Zero123 [26] is also computed under randomly sampled views. The total loss for static SuGaR optimization is:

$$\mathcal{L}_{static} = \lambda^s_{SDS}\mathcal{L}^s_{SDS} + \lambda^s_{ref}\mathcal{L}^s_{ref} + \lambda^s_{mask}\mathcal{L}^s_{mask}, \tag{3}$$

where $\lambda^s_{SDS}$, $\lambda^s_{ref}$ and $\lambda^s_{mask}$ are the weights for different loss terms in static stage.

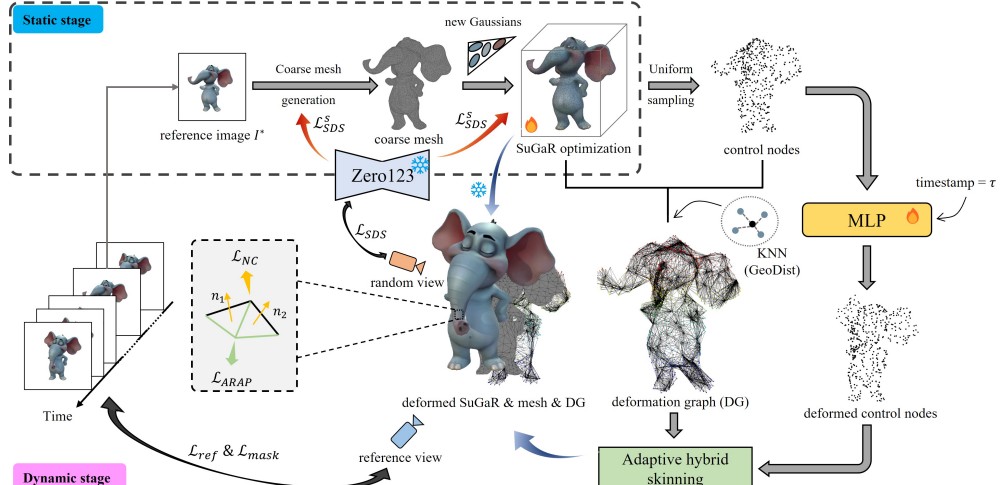

Figure 2: **Overview of DreamMesh4D.** In static stage shown in top left part, a reference image is picked from the input video from with we generate a Gaussian-mesh hybrid representation through a image-to-3D pipeline. As for dynamic stage, we build a deformation graph between mesh vertices and sparse control nodes, and then the mesh and surface Gaussians are deformed by fusing the deformation of control nodes predicted by a MLP through a novel adaptive hybrid skinning algorithm.

## 3.3 Dynamic Stage

In this section, we are going to delve into the deformation procedure for the Gaussian-mesh hybrid representation. First, we will discuss the construction of the deformation graph on the surface of the refined mesh. Then, we will explain the deformation flow, which progresses from the sparse control nodes to the mesh vertices, and ultimately to the surface Gaussians. We will break down each step to give a clear picture of the entire process.

### 3.3.1 Deformation Graph Construction

We begin by uniformly sampling $N_{node}$ sparse points on the surface of the refined mesh to serve as control nodes. To establish connections between the mesh vertices and the sparse control nodes, instead of using simple Euclidean distance to locate the nearest nodes (KNN), we pick $N_{neighbor}$ nodes with the shortest geodesic distance (as indicated by GeoDist in Fig. 2) based on the mesh's topology. This ensures that inappropriate connections between disparate mesh regions are avoided. Then, for a vertex $\mathbf{v}$, the influence $w_{\mathbf{p}}$ of each neighboring control node $\mathbf{p} \in \mathcal{N}(\mathbf{v})$ is calculated following [47]:

$$w_{\mathbf{p}} = \frac{\hat{w}_{\mathbf{p}}}{\sum_{\mathbf{p}_k \in \mathcal{N}(\mathbf{v})} \hat{w}_{\mathbf{p}_k}}, \text{ where } \hat{w}_{\mathbf{p}_k} = \left(1 - \frac{||\mathbf{v} - \mathbf{p}_k||}{d_{\max}}\right)^2, \tag{4}$$

where $d_{\max}$ is the distance to the $(N_{neighbor} + 1)$-nearest node.

### 3.3.2 Deformation with Adaptive Hybrid Skinning

Given that the object's texture was refined in the previous static stage, we fix the Gaussians' appearance properties (color and opacity) and focus the dynamic learning phase solely on spatial properties (positions, rotations, scalings). Initially, the local deformations of the control nodes are predicted by a deformation network $\Psi$. The updated spatial properties post-deformation are subsequently computed by integrating the local deformations of the control nodes through geometric skinning. In particular, given a control node $\mathbf{p} \in \mathcal{P}$ and timestamp $\tau$, the deformation network predict its local deformation at $\tau$ following the equation below. Note that we omit the subscript "$\tau$" is omitted here and in rest paragraphs for simplicity.

$$(R_{\mathbf{p}}, S_{\mathbf{p}}, t_{\mathbf{p}}, \eta_{\mathbf{p}}) = \Psi(\mathbf{p}), \tag{5}$$

where $R_{\mathbf{p}}, S_{\mathbf{p}} \in \mathbb{R}^{3 \times 3}$ and $t_{\mathbf{p}} \in \mathbb{R}^3$ are the rotation, shear matrix and translation respectively. Furthermore, to mitigate artifacts associated with LBS and DQS, we propose an adaptive fusion of their effects to calculate the deformation of mesh vertices. Here, $\eta_{\mathbf{p}} \in \mathbb{R}$ denotes the local rigid strength, indicating the extent to which the region around $\mathbf{p}$ is influenced by DQS at time $\tau$.

**Deformation of Control Nodes**   The predicted shear matrix $S_{\mathbf{p}}$ for node $\mathbf{p}$ is intended only for LBS, whose strength is weaken by the predicted factor $\eta_{\mathbf{p}}$, and the final shear matrix at this location is computed as:

$$\bar{S}_{\mathbf{p}} = (1 - \eta_{\mathbf{p}})S_{\mathbf{p}} + \eta_{\mathbf{p}}\mathbf{I}, \tag{6}$$

where $\mathbf{I} \in \mathbb{R}^{3 \times 3}$ represents an identity matrix indicating no strain effect. Afterwards, the new position of node $\mathbf{p}$ at timestamp $\tau$ is:

$$\tilde{\mathbf{p}} = F_{\mathbf{p}}\mathbf{p} + t_{\mathbf{p}} = R_{\mathbf{p}}\bar{S}_{\mathbf{p}}\mathbf{p} + t_{\mathbf{p}}. \tag{7}$$

**Deformation of Mesh Vertices**   For the deformation of a specific vertex $\mathbf{v}$ using hybrid skinning, the new vertex position calculated with LBS, $\tilde{\mathbf{v}}^{lbs}$, and that with DQS, $\tilde{\mathbf{v}}^{dqs}$, can be obtained following Eq.1 and Eq.2 respectively. The local rigid strength at $\mathbf{v}$ can be computed by linearly blending that of neighboring nodes:

$$\eta_{\mathbf{v}} = \sum_{\mathbf{p} \in \mathcal{N}(\mathbf{v})} w_{\mathbf{p}} \cdot \eta_{\mathbf{p}}, \tag{8}$$

then the fused position of $\mathbf{v}$ after deformation is the interpolation between $\tilde{\mathbf{v}}^{lbs}$ and $\tilde{\mathbf{v}}^{dqs}$:

$$\tilde{\mathbf{v}} = (1 - \eta_{\mathbf{v}})\tilde{\mathbf{v}}^{lbs} + \eta_{\mathbf{v}}\tilde{\mathbf{v}}^{dqs}. \tag{9}$$

The local rotation and strain at $\mathbf{v}$ are the corresponding linear blending from neighboring control nodes:

$$R_{\mathbf{v}} = \exp\left( \sum_{\mathbf{p} \in \mathcal{N}(\mathbf{v})} w_{\mathbf{p}} \cdot \log R_{\mathbf{p}} \right), \tag{10}$$

$$S_{\mathbf{v}} = \sum_{\mathbf{p} \in \mathcal{N}(\mathbf{v})} w_{\mathbf{p}} \cdot \bar{S}_{\mathbf{p}}. \tag{11}$$

**Deformation of Surface Gaussians**   Now we have obtained deformations on the level of mesh vertices, afterwards the deformation on each Gaussian kernel will be calculated. Given a Gaussian kernel $g \in \mathcal{G}(f)$ on a triangle face $f = (\mathbf{v}_a, \mathbf{v}_b, \mathbf{v}_c) \in \mathcal{F}$, its new center at timestamp $\tau$ is computed as:

$$\tilde{\mu}_g = \pi_a \tilde{\mathbf{v}}_a + \pi_b \tilde{\mathbf{v}}_b + \pi_c \tilde{\mathbf{v}}_c, \tag{12}$$

where $(\pi_a, \pi_b, \pi_c)$ is the Gaussian's barycentric coordinate relative to the three triangle vertices. The new rotation is calculated by applying the local rotation $\Delta q_g$ fused from related vertices to its original rotation $q_g$:

$$\Delta q_g = \exp(\pi_a \cdot \log R_{\mathbf{v}_a} + \pi_b \cdot \log R_{\mathbf{v}_b} + \pi_c \cdot \log R_{\mathbf{v}_c}), \tag{13}$$

$$\tilde{q}_g = \Delta q_g \cdot q_g. \tag{14}$$

We further apply the local shear matrix $\Delta S_g$ fused from all three vertices to the original Gaussian scaling $s_g$ to obtain the new scaling:

$$\Delta S_g = \pi_a S_{\mathbf{v_a}} + \pi_b S_{\mathbf{v_b}} + \pi_c S_{\mathbf{v_c}}, \tag{15}$$

$$\tilde{s}_g = \Delta S_g s_g. \tag{16}$$

**Training Losses**   After obtaining the deformed hybrid mesh at timestamp $\tau$, we render its RGB $\hat{I}_\tau^*$ and alpha $\hat{M}_\tau^*$ under reference view. We then compute the reconstruction loss $\mathcal{L}_{ref} = ||\hat{I}_\tau^* - I_\tau^*||_2^2$ and mask loss $\mathcal{L}_{mask} = ||\hat{M}_\tau^* - M_\tau^*||_2^2$, where $I_\tau^*$ and $M_\tau^*$ are the ground truth image and mask of input video at timestamp $\tau$. For supervision under other views, we calculate SDS loss $\mathcal{L}_{SDS}$ based on Zero123 [26] under randomly sampled views. Furthermore, our mesh-based representation naturally facilitates the introduction of local rigid constraints by leveraging the topology of the mesh. Specifically, the as-rigid-as-possible (ARAP) loss [46] is computed as:

$$\mathcal{L}_{ARAP} = \sum_{\mathbf{v} \in \mathcal{V}} \sum_{\mathbf{v}_n \in \mathcal{C}(\mathbf{v})} \omega_n(\mathbf{v}) \big|\big|(\tilde{\mathbf{v}} - \tilde{\mathbf{v}}_n) - R_{\mathbf{v}}(\mathbf{v} - \mathbf{v}_n)\big|\big|_2^2, \tag{17}$$

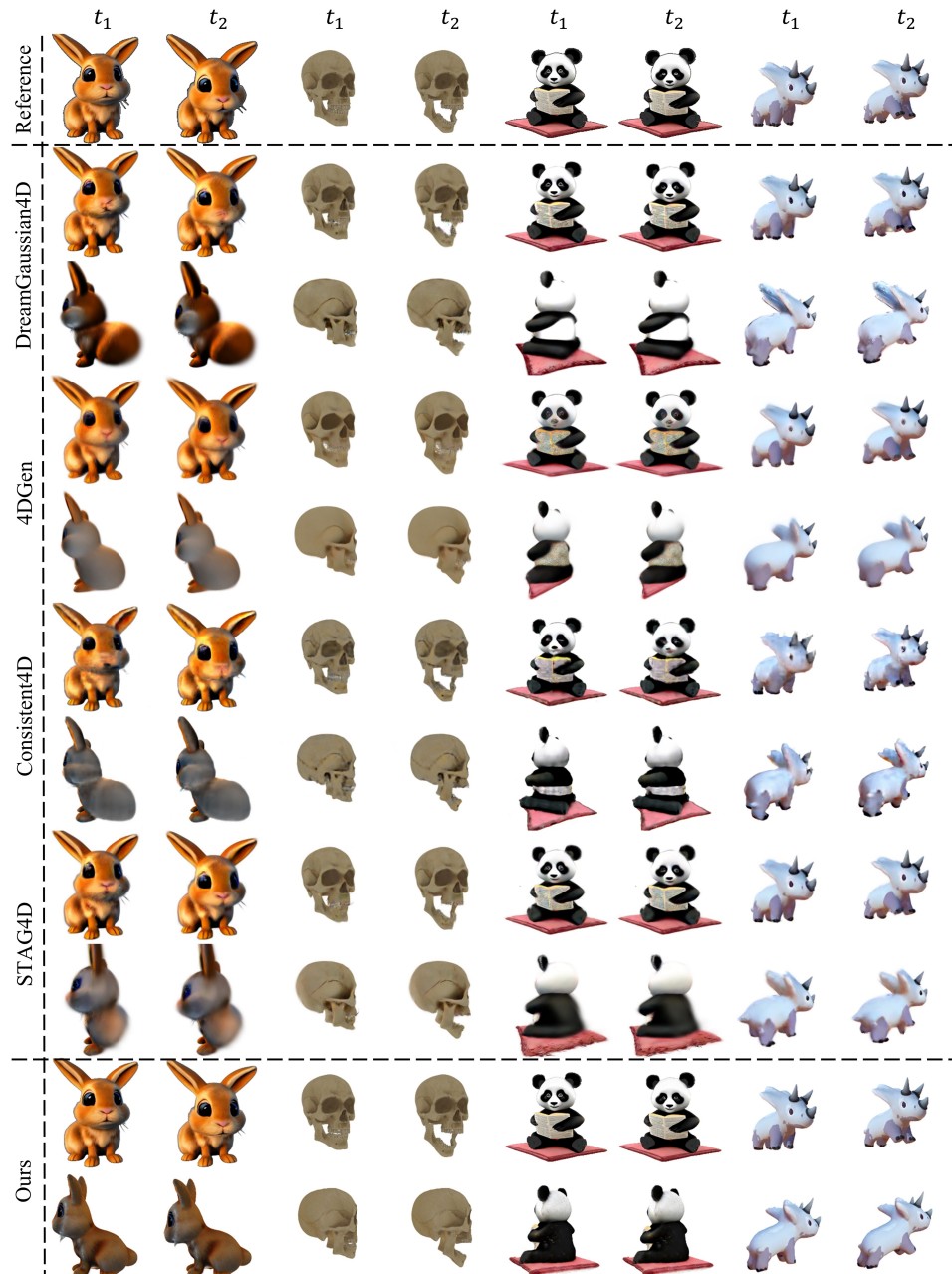

Figure 3: **Qualitative comparison with baselines.** We compare our method with 4 previous video-to-4D methods. The first row provides two ground truth frames for each case. For each compared method, we render each case under reference view and another novel view at the two timestamps. The result demonstrates that our method is able to generate sharper 4D content with rich details, especially for the novel views. Please zoom in for more details.

where $\mathcal{C}(\mathbf{v})$ represents the one-ring neighbors of vertex $\mathbf{v}$, and $\omega_n(\mathbf{v})$ is the cotangent weight [29] between $\mathbf{v}$ and its $n$-th connected vertex $\mathbf{v}_n$. Furthermore, we also introduce the normal consistency loss $\mathcal{L}_{NC}$ provided by PyTorch3D [37] on the deformed mesh to globally constrain the mesh surface. Hence, the overall objective for our dynamic stage is a weighted combination of the above loss terms:

$$\mathcal{L}_{dynamic} = \lambda_{SDS}\mathcal{L}_{SDS} + \lambda_{ref}\mathcal{L}_{ref} + \lambda_{mask}\mathcal{L}_{mask} + \lambda_{ARAP}\mathcal{L}_{ARAP} + \lambda_{NC}\mathcal{L}_{NC}, \quad (18)$$

where $\lambda_{SDS}$, $\lambda_{ref}$, $\lambda_{mask}$, $\lambda_{ARAP}$ and $\lambda_{NC}$ are strengths of different loss terms in dynamic optimization stage.

| | PSNR(ref) ↑ | SSIM(ref) ↑ | LPIPS ↓ | FVD ↓ | FID-VID ↓ | CLIP ↑ |
|---|---|---|---|---|---|---|
| Consistent4D | 26.58 | 0.935 | 0.133 | 929.39 | 31.84 | 0.917 |
| DreamGaussian4D | 31.06 | 0.947 | 0.143 | 994.11 | 32.73 | 0.913 |
| 4DGen (16 frames) | 27.02 | 0.937 | 0.137 | 913.10 | 63.32 | 0.909 |
| STAG4D | 27.99 | 0.941 | 0.136 | 1048.10 | 38.77 | 0.905 |
| Ours | **37.04** | **0.980** | **0.126** | **474.96** | **29.14** | **0.938** |

Table 1: **Quantitative comparison with baselines.** Our method achieves best score on all metrics.

## 4 Experiments

### 4.1 Experiment Setup

**Dataset**: Our quantitative results are evaluated on the test dataset provided by Consistent4D [12], which contains seven multi-view videos. Each video has one input view for scene generation and four testing views for evaluation. For qualitative evaluation, we curate a set of challenging videos from previous works [12] and those generated by the video diffusion model SVD [3]. **Evaluation metrics**: The per-frame LPIPS [67] score and the CLIP-score [36] are computed between the testing and rendered videos, with the final scores averaged over the four testing views. These two scores serve as image-level metric to assess the structural and semantic similarity between the rendered images and the ground truth. Furthermore, we compute the FID-VID [2] and FVD [52] as video-level metrics to evaluate the video temporal coherence. Note that we report PSNR and SSIM values only for the reference view, as pixel- and patch-wise similarities are too sensitive to reconstruction differences, making them unsuitable for evaluating novel views in our generation task. However, we find they are suitable for evaluating the method's ability of modeling sharp features in the reference view. **Baselines**: We compare our method with previous video-to-4D generation methods: Consistent4D [12], DreamGaussian4D [38], 4DGen [65] and STAG4D [66]. All the experiments of above baselines are conducted using the code from their official GitHub repository.

### 4.2 Comparison

**Qualitative Comparison**  Fig. 3 shows qualitative results of our method compared to other baseline works. The results reveal that our method generates 4D objects with higher fidelity and more details under reference view. And our method also outperforms other works with better spatial-temporal consistency, demonstrating the effectiveness of our method. Please zoom in for more details.

**Quantitative Comparison**  Table 1 demonstrates superior performance of our method against other baseline works quantitatively. Specifically, our approach notably exceeds the existing state-of-the-art in all measured metrics. Our method excels in both PSNR and SSIM, indicating a high level of reconstruction accuracy. Furthermore, the FVD score is particularly impressive, being only half that of competing methods. We also achieve the lowest FID-VID score, suggesting a significant enhancement in video quality produced by our 4D generation technique. Finally, our method achieves the lowest LPIPS and highest CLIP scores, ensuring both high image realism and semantic consistency. Overall, the numerical data clearly demonstrate the superior capabilities of our method in translating video to 4D content.

### 4.3 Ablation Study

In this section, we conduct ablation study to analyse the impact of various components on the performance of our method. The components under consideration include: (a) the choice between Euclidean and geodesic distance (EucDist and GeoDist) when constructing deformation graph; (b) our proposed adaptive hybrid skinning algorithm; (c) the ARAP and normal consistency terms (geometric regularization terms); (d) the choice between vanilla 3D Gaussians [18] and Gaussian-mesh representation for our base 3D representation.

**EucDist vs. GeoDist**  Fig. 4(a) provides a comprehensive qualitative analysis of the choice between EucDist and GeoDist. When using EucDist, the vertices on the elephant are incorrectly connected

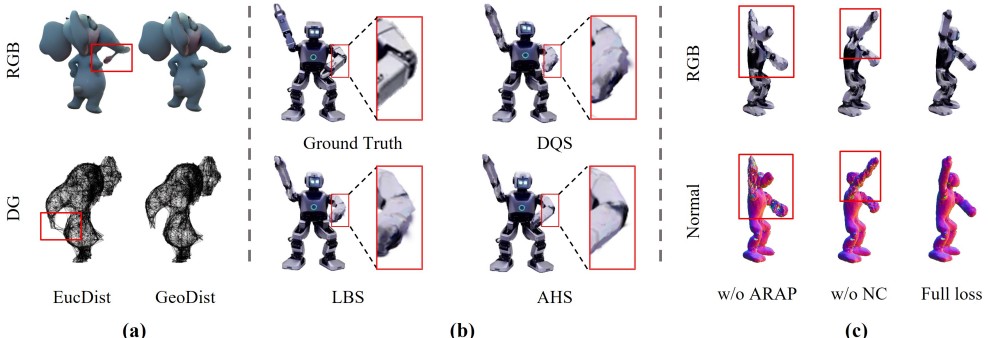

**(a)**                  **(b)**                 **(c)**

Figure 4: **Qualitative evaluation of ablation studies on:** (a) choice between GeoDist and EucDist for deformation graph (DG) construction; (b) our proposed adaptive hybrid skinning (AHS) against LBS and DQS; (c) effects of ARAP and normal consistency (NC) loss.

| | PSNR(ref) ↑ | SSIM(ref) ↑ | LPIPS ↓ | FVD ↓ | FID-VID ↓ | CLIP ↑ |
|---|---|---|---|---|---|---|
| Skinning=DQS | 36.28 | 0.978 | 0.126 | 479.83 | 29.78 | **0.940** |
| Skinning=LBS | 36.68 | 0.980 | 0.155 | 540.86 | 32.19 | 0.928 |
| w/o arap | 37.04 | 0.979 | 0.142 | 751.56 | 42.08 | 0.907 |
| w/o normal consistency | 36.86 | 0.980 | 0.147 | 519.49 | 30.90 | 0.932 |
| Full method | **37.04** | **0.980** | **0.126** | **474.96** | **29.14** | 0.938 |

Table 2: **Quantitative evaluation of ablation study on different components.** During experiments, we keep all other setup unchanged compared to the full method except the tested components. The quantitative scores show that our full method achieves best performance on almost all metrics.

to its body, resulting in significant deformation artifacts. Conversely, GeoDist correctly links the vertices to neighboring nodes, enabling smooth object motion.

**Adaptive Hybrid Skinning**    The upper two rows of Table 2 presents quantitative results when replacing our adaptive hybrid skinning with DQS and LBS respectively. Almost all metrics have a decrease compared to using our adaptive hybrid skinning, showcasing its robustness. We also provide a qualitative comparison on a dancing robot case in Fig. 4(b). When LBS or DQS is used, there are artifacts on the robot's deformed elbow along with uneven surface. In contrast, the artifacts are eliminated and the surface becomes smooth when our adaptive hybrid skinning is used.

**Geometric Regularization Terms**    The third and fourth rows of Table 2 present the scores of metrics when either the ARAP or normal consistency term, respectively, is omitted from Equation 18. Decreases are observed across all metrics compared to full loss terms and it drops significantly when disabling ARAP loss. This circumstance matches the qualitative analysis shown in Fig. 4(c). Without ARAP term, it appears serious distortions on the object geometry. When the normal consistency term is disabled, the object surface becomes less smooth, and consequently, the texture is impaired.

**3D Gaussians vs. Gaussian-mesh Hybrid**    In Fig. 5, we provide the qualitative comparison between 3D Gaussians and the Gaussian-mesh hybrid representation for our base 3D representation. It shows that when utilizing 3D Gaussians, the texture of generated objects is blurry on those parts unseen in reference image. As a contrary, Gaussian-mesh hybrid representation presents clean and high-quality texture under every view, which benefits from the sharp surface provided by mesh.

### 4.4 Limitations

Despite the superior results achieved, our work still exist several limitations. In particular, our method relies on a pre-trained multi-view diffusion model (Zero123) for novel view supervision through SDS, leading to long optimization time and limited performance on which case Zero123 cannot handle well. Moreover, our method is currently only designed to generate 4D contents at object level from input

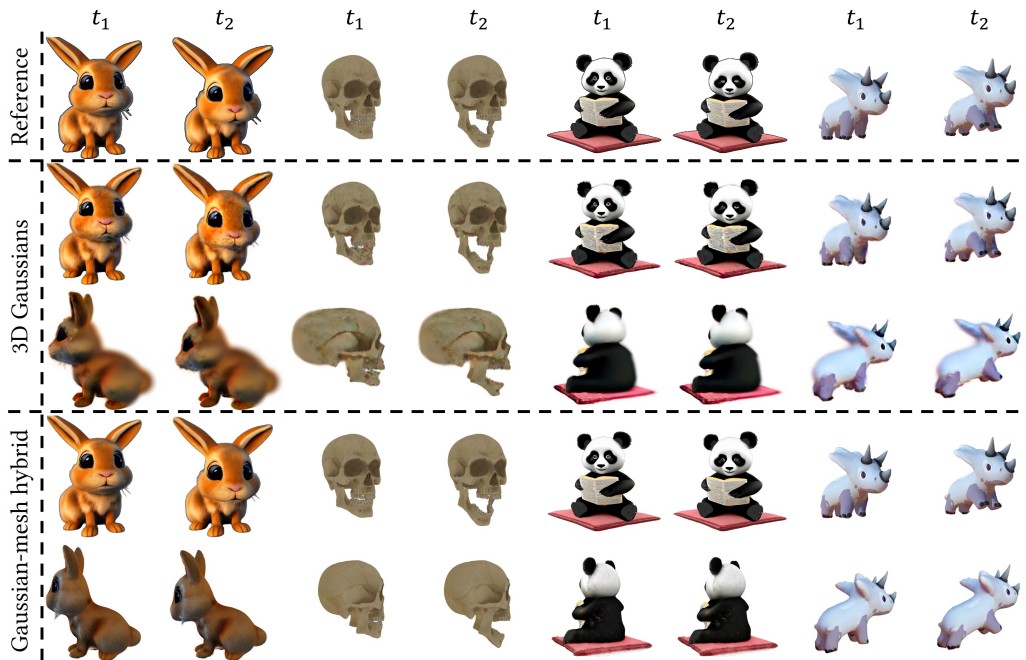

Figure 5: **Qualitative comparison on 3D representation between 3D Gaussians and Gaussian-mesh hybrid representation.** When utilizing 3D Gaussians as our base 3D representation, the texture is blurry on the parts unseen in reference image. As a comparison, the texture is clean and of high quality under every view when employing the Gaussian-mesh hybrid representation.

videos captured under fixed viewpoint. The extension of our framework to scene-level generation or to videos captured with a moving camera remains an area for future exploration. Finally, due to the scarcity of test data, the performance of our method on more complex task is not evaluated. These identified limitations will be addressed in our future research endeavors.

## 5 Conclusion

In this work, we introduce DreamMesh4D, an innovative video-to-4D framework that generates dynamic meshes through a static-to-dynamic optimization process. By employing a Gaussian-mesh hybrid representation, we simultaneously refine both the geometry and texture of the object. This approach allows the static object to serve as an excellent starting point for dynamic learning. During the dynamic stage, we construct a deformation graph on the object's surface using geodesic distance. Thereafter, the motion of the entire mesh, as well as the surface Gaussians, are driven by sparse control nodes via a novel geometric skinning algorithm named adaptive hybrid skinning. It benefits from the strengths of both Linear Blending Skinning (LBS) and Dual-Quaternion Skinning (DQS), enabling more robust deformation. Extensive experiments have demonstrated the superior performance of our method in generating high-fidelity 4D objects. It significantly surpasses previous methods in both rendering quality and spatial-temporal consistency, establishing a new benchmark for video-to-4D tasks. While our method benefits a lot from the mesh-based representation, it reveals a promising direction in the field of video-to-4D generation. Furthermore, our method's compatibility with modern geometric pipelines showcases its potential applicability in the 3D gaming and film industries.

## Acknowledgement

This work was supported in part by National Science and Technology Major Project (2022ZD0115102), in part by National Natural Science Foundation of China (62202389), in part by a grant from the Westlake University-Muyuan Joint Research Institute, and in part by the Westlake Education Foundation.

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

# A    Appendix / supplemental material

## A.1    Additional Qualitative Results

Here we provide more qualitative comparisons of our method against baseline works in Fig. 6. Please see our project page for more temporal qualitative results with video format.

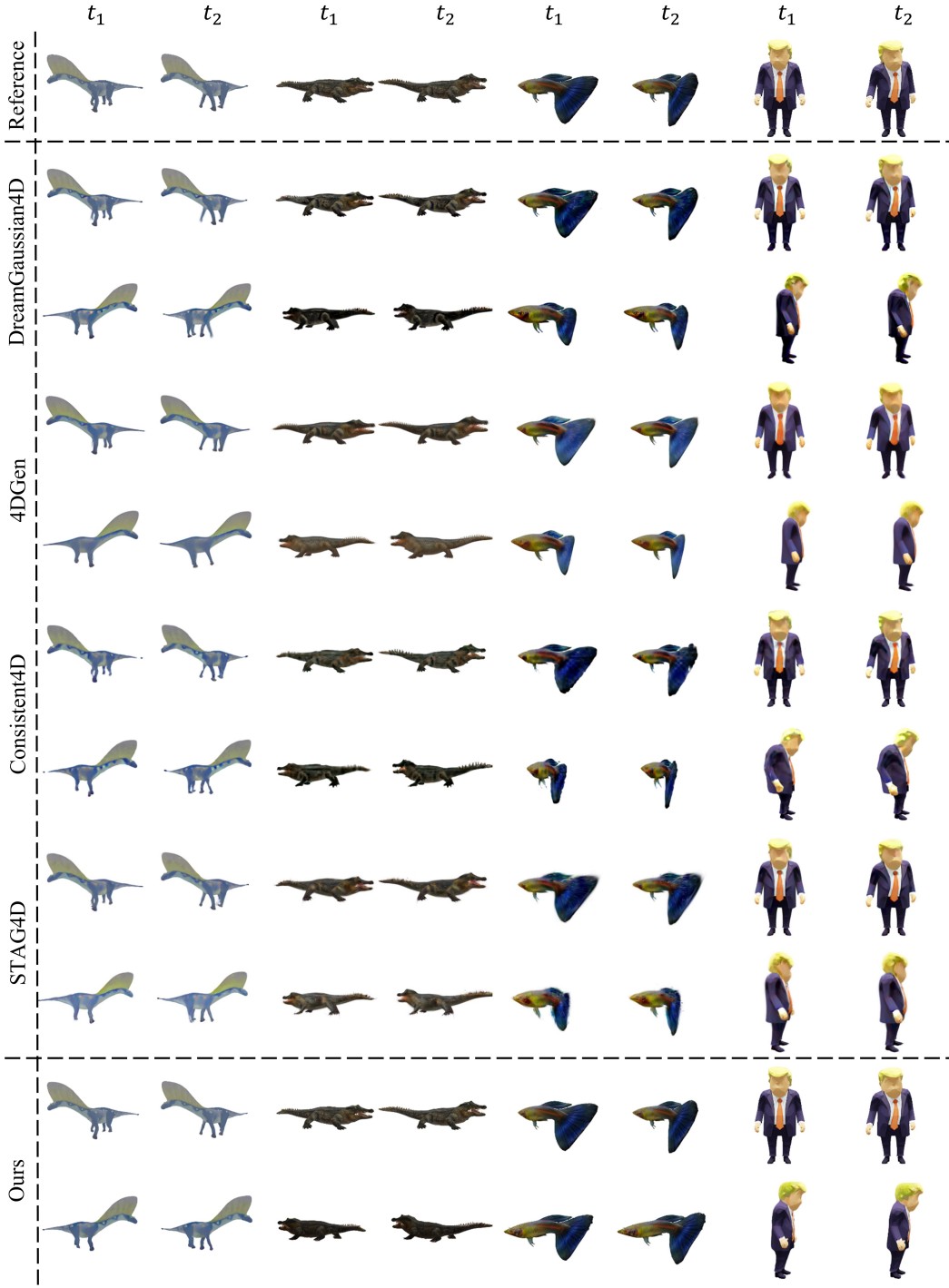

Figure 6: **Additional qualitative comparison with baselines.**

## A.2 Implementation Details

**Static Stage**  In the process of both coarse mesh generation and SuGaR refinement, the loss in Equation 3 is utilized for supervision, with the strength of different terms as $\lambda_{SDS}^s = 0.1$, $\lambda_{ref}^s = 1000$ and $\lambda_{mask}^s = 500$. For the generation of coarse mesh, a set of randomly initialized 3D Gaussians are optimized for 3000 steps in total. In the former 1500 steps, we do densification and pruning every 100 steps. After the 1500 steps, densification and pruning are stopped, and we introduce additional opacity binary and density regularization terms as described in [9] into optimization until step 3000. Finally, we prune all Gaussians with opacity less than 0.5 and extract coarse mesh through Poisson reconstruction. Afterwards, we bind $x = 6$ new flat Gaussians to each triangle face of the coarse mesh, and do optimization for 2000 steps.

**Dynamic Stage**  In dynamic stage, we defaultly sample $N_{node} = 1024$ control nodes and assign $N_{neighbor} = 4$ neighboring nodes for each vertex when constructing deformation graph. For each training step, 8 frames are randomly sampled from the input video for supervision. And for each sampled timestamp, we randomly sample 2 views for the calculation of SDS loss. All images are rendered at resolution $512 \times 512$ with white background. The camera distance to world center is fixed as 3.8 and the degree of field-of-view (FoV) is fixed as $20°$. As for the strenghts of different loss terms, we defaulty set $\lambda_{SDS} = 0.1$, $\lambda_{ref} = 5000$, $\lambda_{mask} = 500$ and $\lambda_{NC} = 10$. The value of $\lambda_{ARAP}$ is chosen case-specifically in [1, 10] according to the motion amplitude of object. The deformation network is zero-ly initialized and totally optimized for 2000 steps with learning rate as 0.00032. All of our experiments are conducted on a single NVIDIA RTX 4090 GPU.

**Licenses**  Here we provide the URL, citations, and licenses of open-sourced assets used in this work in Table 3.

| URL | Citation | License |
|---|---|---|
| https://github.com/threestudio-project/threestudio | [10] | Apache-2.0 license |
| https://github.com/huggingface/diffusers | [53] | Apache-2.0 license |
| https://github.com/facebookresearch/pytorch3d | [37] | BSD License |
| https://github.com/cvlab-columbia/zero123 | [26] | MIT License |
| https://github.com/Anttwo/SuGaR | [9] | Gaussian-Splatting License |

Table 3: **URL, citations and licenses of the open-sourced assets used in this work.**

## A.3 Additional Experiments

| | Consistent4D | DreamGaussian4D | 4DGen | STAG4D | Ours |
|---|---|---|---|---|---|
| Training Time | 2.0h | 0.6h | 3.0h | 1.6h | 0.8h |
| Memory | 28GB | 20GB | 15GB | 7GB | 8GB |

Table 4: Comparison of computation cost of different methods.

**Computation Cost**  In Table 4, we report the computation cost of our method and other compared baseline methods, demonstrating the computation efficiency of our method.

| #Gaussians per face | PSNR(ref)↑ | SSIM(ref)↑ | LPIPS↓ | FVD↓ | FID-VID↓ | CLIP↑ |
|---|---|---|---|---|---|---|
| 1 | 36.17 | 0.977 | 0.134 | 523.39 | 27.18 | 0.939 |
| 3 | 36.55 | 0.979 | 0.129 | 496.46 | 27.60 | **0.943** |
| 4 | 36.60 | 0.979 | 0.128 | 519.35 | 26.81 | 0.940 |
| 6 | **36.63** | **0.979** | **0.127** | **477.63** | **25.96** | 0.940 |

Table 5: Quantitative evaluation of ablation study on the number of Gaussians per triangle face.

**Ablation on Number of Face Gaussians**  We also conduct comparisons on different number of Gaussians per face. The qualitative and quantitative comparison results are presented in Fig. 7 and Table A.3. The results demonstrate that the more Gaussains utilized per triangle face, the more detailed appearance can be obtained (e.g., the eyes and nose of the dog). Empirically we find that 6-Gaussians-per-face can already deliver satisfying performance by considering the rendering quality and training time. Hence we keep this setup for all experiment cases.

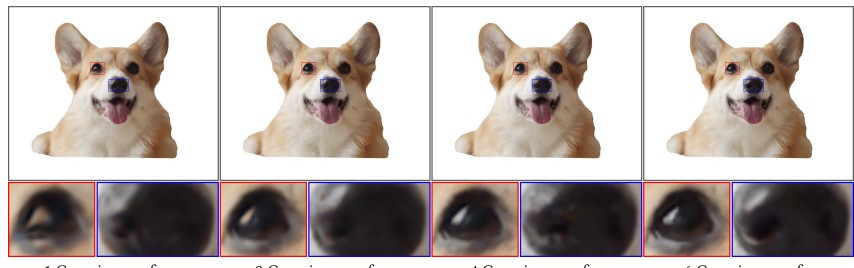

| 1 Gaussian per face | 3 Gaussians per face | 4 Gaussians per face | 6 Gaussians per face |

Figure 7: **Qualitative comparison on the number of Gaussians per face.** The appearance quality of details (e.g. the eyes and nose) is getting better when binding more number of Gaussians on triangle face.

**Ablation on Number of Control Nodes** Here we conduct an additional ablation study on the choice of the number of control nodes, $N_{node}$, and the number of connected nodes for each vertex, $N_{neighbor}$. We try [256, 512, 1024] for $N_{node}$ and [4, 8, 16] for $N_{neighbor}$, and the quantitative results are presented in Table 6. There are no significant distinct on scores of different metrics under different combination of $N_{node}$ and $N_{neighbor}$, except for PSNR(ref), on which $\{N_{node} = 1024, N_{neighbor} = 4\}$ achieves the highest score. While PSNR(ref) is a key metric revealing reconstruction quality, we pick $\{N_{node} = 1024, N_{neighbor} = 4\}$ as the default setup for our method.

| | PSNR(ref) | SSIM(ref) | LPIPS | FVD | FID-VID | CLIP |
|---|---|---|---|---|---|---|
| $N_{node} = 1024, N_{neighbor} = 16$ | 36.39 | 0.979 | 0.128 | 559.43 | 32.08 | 0.932 |
| $N_{node} = 1024, N_{neighbor} = 8$ | 36.70 | 0.980 | 0.128 | 521.15 | 30.94 | 0.936 |
| $N_{node} = 1024, N_{neighbor} = 4$ | **37.82** | 0.980 | 0.128 | 516.46 | 31.20 | 0.937 |
| $N_{node} = 512, N_{neighbor} = 16$ | 36.14 | 0.979 | 0.127 | 542.32 | **30.19** | 0.937 |
| $N_{node} = 512, N_{neighbor} = 8$ | 36.28 | 0.979 | 0.127 | **505.86** | 30.34 | 0.935 |
| $N_{node} = 512, N_{neighbor} = 4$ | 36.42 | 0.979 | 0.127 | 511.78 | 30.59 | 0.936 |
| $N_{node} = 256, N_{neighbor} = 8$ | 35.92 | 0.978 | 0.129 | 512.25 | 32.21 | 0.935 |
| $N_{node} = 256, N_{neighbor} = 4$ | 35.97 | 0.979 | 0.127 | 522.46 | 30.55 | 0.935 |

Table 6: **Quantitative results of setup on number of control nodes and mesh vertex connectivity.**

