# OpenReview forum: "DreamMesh4D: Video-to-4D Generation with Sparse-Controlled Gaussian-Mesh Hybrid Representation"
_NeurIPS.cc/2024/Conference — NeurIPS 2024 poster_

### Official Review · Reviewer_2HE6 · 2024-06-26

**Soundness:** 3
**Presentation:** 3
**Contribution:** 3
**Rating:** 7
**Confidence:** 3

**Summary:**

The paper proposes a framework to generate animated surfaces from input videos. It has a static step to generate a base triangle mesh, and textures represented by 3D Gaussians. A subsequent dynamic step changes the positions of the mesh vertices and the gaussians. The framework leverages on Zero123 and SuGaR to generate the mesh and gaussians, integrating loss terms from those approaches in its training mechanism. Tha dynamic step uses geodesic distances to create a deformation graph and include As-Rigid-As-Possible and Normal Consistency loss terms.

**Strengths:**

(1) This paper is very informative and well contextualized. I am a specialist in geometry but not in Diffusion models, but I was able to use the Introduction and Related Work sections to familiarize myself with the theme. That was very fun. There are lots of the references from very recent papers, so good job on keeping track of the field!

(2) The results are probably ready for current Computer Graphics pipelines, since it uses meshes and skinning.

(3) I liked the mathematical modeling. Explaining everything in terms of the losses made a good paper structure. I have reservations about the specific mathematical notation (see Weaknesses), but reading was good overall.

(4) I liked the proposed method. All losses terms make sense and are intuitive. Knowing that the SuGaR SDS would work well in a more complex pipeline was informative.

(5) I liked the use of geodesic distance to compute the deformation graph. It is more robust than euclidean distance indeed.

**Weaknesses:**

(1) I missed a video in the supplementary material. As a 4D approach, the paper would benefit greatly from showing one. It shows metrics about temporal coherence, but a video would avoid any possible doubt.

(2) I'm missing references about 4D neural implicit surfaces in the 4D Representations paragraph in the Related Work section. They also deal with dynamic surfaces in neural pipelines. I believe a sentence including all of them would be sufficient.

Novello, Tiago, et al. "Neural Implicit Surface Evolution." Proceedings of the IEEE/CVF International Conference on Computer Vision. 2023.

Guandao Yang, Serge Belongie, Bharath Hariharan, and Vladlen Koltun. Geometry processing with neural fields. Advances in Neural Information Processing Systems, 34, 2021.

Mehta, Ishit, Manmohan Chandraker, and Ravi Ramamoorthi. "A level set theory for neural implicit evolution under explicit flows." European Conference on Computer Vision. Cham: Springer Nature Switzerland, 2022.

(3) The mathematical notation is a little bit polluted, impacting reading in my opinion. Subscripts and superscripts are used too much. The text would be cleaner by using more symbol letters instead. A good rule of thumb is to avoid using subscripts and superscripts unless they represent indices.

The same letters are also used for very different contexts. N is used for number of vertices and control point set, for example. Even though different fonts are used in each context, the cognition when reading is to remember N and forget the font used. It would be cleaner to use different symbols.

**Questions:**

(1) I would like to know more about the temporal coherence in the proposed approach, since a video is not available. Even though the paper show metrics in that respect I would also like to see qualitative results.

(2) Even though the results are probably ready for a CG pipeline, I believe the training and inference must be computationally expensive. I would like to know the training and inference times, and the hardware used.

(3) Does only changing positions, rotations, and scaling of the gaussians in the deformation step result in accurate view-dependent effects? In theory the spherical harmonics should also be recalculated.

(4) Why 6 gaussians per triangle in the static stage?

(5) Equation 10 should come with a small explanation to give intuition. Mentioning that the logarithmic rotation is a mapping to the Lie Algebra and that the exp is a conversion back to the rotation matrix form would at least give a direction for a reader that is not a specialist.

(6) I believe the references in line 38 [29, 54, 17, 41, 9] are not associated directly with score distillation sampling. Is that a typo?

**Limitations:**

Yes, the paper has a specific section describing limitations.

---

> ### Author Rebuttal · Authors · 2024-08-06
>
> The authors are grateful for the insightful feedback and support from the reviewer. Below we address the mentioned concerns separately.
>
> ### **Q1: Video qualitative results.**
> **A**: In the first paragraph of appendix, we have provided the link to our anonymous project page, which contains the input video and our rendered video results as well as our blender demo. Please check it.
>
> ### **Q4: Concerns about computation cost of training and inference**
> **A**: For inference, which is the process of computing the deformed 3D object under a given timestamp, we report the time consuming of different methods as below:
> ||Consistent4D|DreamGaussian4D|4DGen|STAG4D|Ours|
> |:---:|:---:|:---:|:---:|:---:|:---:|
> |Inference Time|51ms|3ms|3ms|3ms|3ms(MLP)+11ms(Skinning)|
>
> Among these methods, Consistent4D utilizes implicit representation and volume rendering, leading to much more query points for the deformation network and hence the longest inference time (51ms). Compared to those Gaussian-based methods, DreamGaussian4D, 4DGen and STAG4D, our method has a little bit slower inference time (14ms). The extra computation overhead comes from the skinning calculation, which is implemented with pure PyTorch. This overhead can be eliminated through some engineering effort of implementing it as a CUDA extension, and we leave it for future works.
>
> As for the computation cost of training, we have reported the overall training time and GPU memory usage in our global rebuttal. Compared to other methods, our method is the most computational efficient considering both the time and memory consuming (0.8h and 8GB).
>
> ### **Q5: Concerns on updating spherical harmonics.**
> **A**: Following previous 3D generation works on 3D Gaussians, we use 0-degree spherical harmonics which is actually equivalent to RGB color without view-dependent effects. Hence we do not compute the deformation of spherical harmonics. We would explore to use higher level SH-coefficients for view-dependent effects in future.
>
> ### **Q6: Why 6 Gaussians per triangle in the static stage?**
> **A**: By compromising on the appearance representation capability and computational cost, we choose 6 Gaussians per triangle. More Gaussians would enable a better capability to represent the surface appearance, with an expense of more computational cost. As shown in Figure 2 and Table 3 of the supplemented PDF rebuttal, the experimental results demonstrate 6-Gaussians can already deliver satisfying performance by considering the rendering quality and training time.
> It is worth mentioning that, even with only 1 Gaussians per face, our method still largely outperforms previous methods with the fastest training time.
>
> ### **Q2\&Q3\&Q7\&Q8: Concerns about writing.**
> **A**: We really appreciate your advice on the paper writing, and we will revise the manuscript according to the constructive suggestions. We will include the works on 4D neural implicit surfaces in the 4D representations paragraph in the Related Work section. All mathematical notations in the paper will be carefully simplified for easier reading. And we will supplement a proper explanation about Equation 10 as you mentioned in Question (5). Finally, the references in line 38 are indeed a typo, and the words before these references should be "differentiable 3D representations" rather than "3D generation".

---

> > ### Comment · Reviewer_2HE6 · 2024-08-07
> >
> > Thank you for the responses. I missed the link for the result videos at the appendix, and I believe they are good enough. There are minor artifacts in the normals, but my concerns about temporal coherence are addressed. I think the paper is very well written and deserves publication, specially for the good contextualization, clean presentation of the theory, and the coherent mathematical treatment of the problem. It would be a very good reference for someone not so familiar with the field.

---

> ### Author Response · Authors · 2024-08-11
>
> The authors are grateful for your support! We will further polish our method in the final version.

---

### Official Review · Reviewer_ow51 · 2024-07-08

**Soundness:** 3
**Presentation:** 3
**Contribution:** 1
**Rating:** 5
**Confidence:** 5

**Summary:**

DreamMesh4D is a novel framework for transforming static images into 4D dynamic meshes. It refines object geometry and texture using a Gaussian-mesh hybrid and a geodesic-based deformation graph. A new skinning algorithm combines the best of LBS and DQS for enhanced deformation. The method excels in creating high-quality 4D objects with improved rendering and consistency.

**Strengths:**

1.The results show an improvement in both quantitative and qualitative metrics compared to previous methods.
2.The use of points to control motion is an interesting approach.
3.The writing is clear.

**Weaknesses:**

1.The novelty of this work is somewhat limited in comparison to previous methods. The training strategy, the structure of the loss function, and the design of deformation are quite similar. The main difference lies in the representation, which is a predictable technical combination.

2.I find that the detail of the results is superior to those of previous methods. Therefore, in section 4.2, the author should elucidate which aspects of the pipeline design have contributed to the improved quality compared to the previous method's design.

3.The paper argues that previous works have significant time and memory complexity, but there is a lack of comparison regarding time and GPU usage costs. It would be beneficial to include such a comparison to demonstrate the efficiency of the proposed method.

**Questions:**

The current article lacks interesting insights. I suggest that the author should focus more on the advantages of this representation and explain why it is important for the 4D generation task.

**Limitations:**

Refer to the Weaknesses and Questions section

---

> ### Author Rebuttal · Authors · 2024-08-06
>
> The authors are grateful for the valuable and in-depth feedback of the reviewer. Below we address the mentioned concerns separately.
>
> ### **Q1: Limited novelty compared to previous methods.**
> **A**: Previous methods for monocular video-to-4D generation include Consistent4D, DreamGaussian4D, 4DGen and STAG4D. All of them directly employ existing dynamic neural radiance fields or 3D Gaussian Splatting as their base representations, and predict query points' deformation with a MLP network. In contrast, we are the first to exploit mesh as the underlying representation for the problem and demonstrate superior effectiveness compared to prior methods. Although it might be predictable, no prior work has really tried and demonstrated the possibility of the idea for this challenging task. As both reviewer 1 and reviewer 3 also agree on, the method to generate dynamic meshes is new and informative. The authors humbly think the proposed method would provide valuable insight for following works.
>
> ### **Q2: Which aspects of the pipeline have contributed to the quality improvement compared to previous methods?**
> **A**: The main aspect contributing to the quality improvement is the employment of mesh representation. 3D mesh models surface explicitly and is crucial for appearance modeling and animation. To validate the claim, we further conduct a thorough ablation study by replacing the mesh representation with no surface bound 3D Gaussian Splatting, and leave other parts unchanged. The experimental results demonstrate the mesh representation achieves a PSNR 36.73 dB compared to 28.71 dB with 3D-GS. It demonstrates the superior advantage by exploiting surface bound Gaussian Splatting for this challenging less constrained optimization problem. Detailed quantitative and qualitative evaluation results can be found in the rebuttal PDF (i.e. Table 2 and Figure 1).
>
> ### **Q3: Concerns about computation cost.**
> **A**: The detailed comparisons against prior works in terms of time and GPU memory usage can be found in the global rebuttal section. We evaluate all the methods on a single NVIDIA RTX 6000Ada graphic card since Consistent4D cannot run on a 24G GPU. Our method requires 0.8h and 8G memory for training, which is superior compared to Consistent4D, DreamGaussian4D, 4DGen and STAG4D. For example, Consistent4D requires 2h and 28G memory for training, STAG4D requires 1.6h and 7G memory for training.
>
> ### **Q4: Insights of this work.**
> **A**: The 4D generation task is inherently not a well constrained problem, usually only a sequence of monocular video is given. Additional constrains should thus be enforced to achieve better performance. Prior works usually rely on a pre-trained image diffusion model to provide extra constrain in the form of score distillation loss. Although they can deliver impressive performance, the quality of the generated 4D asset still has room for improvement, e.g. the asset usually exhibits blurry appearance or unsatisfactory geometry as shown in our experimental results.
>
> The authors suspect the performance gap might be caused by the used loosely constrained 4D representation, i.e. NeRF or 3D-GS with an additional deformation network for dynamic modeling, which is commonly used for multi-view 4D reconstruction problem. Since multi-view cues are usually sufficient to constrain the problem, prior commonly used 4D representations would thus still be able to deliver good performance. However, it is not the case for 4D generation from a monocular video, which provides much less multi-view cues.
>
> Therefore, the authors thought additional constraint should be enforced to better constrain this more challenging problem, which motivates us to explore the usage of the Gaussian-mesh hybrid representation. Gaussian-mesh representation was firstly proposed in SuGaR and used for triangular mesh extraction from a pretrained 3D-GS model. It enforces 3D Gaussians to bind on the object surface and degenerate towards 2D Gaussians. This kind of enforcement reduces the parameter searching space of the optimization problem, and thus could deliver better performance. The experimental results also prove our hypothesis and demonstrate superior performance compared to prior works.
>
> However, it is not trivial to simply integrate Gaussian-mesh representation into a 4D pipeline. For example, 1) how to efficiently and effectively deform the mesh-Gaussian representation for 4D dynamics? 2) how to make sure the surface bound Gaussians are still able to render high-quality textures under deformation? It is those challenges motivating us: 1) to propose the new skinning algorithm combines the best of LBS and DQS for this particular problem; 2) to propose the deformation of surface Gaussian attributes; 3) the usage of deformation graph to provide additional geometric deformation constraints. It is all those factors and insights leading to the final formulation and superior performance of our method.
>
> Furthermore, mesh based representation is already very mature for modern CG pipelines. Generated 4D assets with mesh representation and skinning method can be conveniently exported to existing commercial pipelines, which further motivates us to exploit this representation. The authors humbly think our work would be valuable to the community and provide good insight for following works in relevant fields.

---

> > ### Comment · Reviewer_ow51 · 2024-08-12
> >
> > 1. I also missed the video results like Reviewer 2HE6，because I previously believed that links were not allowed. Though I think the video details are good. However, I am concerned about why the orbital view videos are not shown, as I can't judge the spatial consistency.
> >
> > 2.My concerns are quite similar to those of Reviewer 7V8W. In Q3: Innovative Contribution of this work under 7V8W, I find the author summarized two contributions. Regarding the changing of Gaussian attributes during deformation, I think it is a basic strategy. In 4DGS (CVPR 2024), the positions, rotations, and scalings are also deformed. Regarding the new skinning algorithm, in the ablation study in the main paper, the evaluation metric did not significantly exceed DQS and LBS. So it is not the key. In the rebuttal PDF, I think the ablation about GS and hyper is important.I admit that the representation's blur problems are mitigated, and I believe the improvement contributes to the changes in representation, but the technology is existing. Therefore, I think it is a not bad paper, but not an excellent one. So I do not rate it above "boardline accept."

---

> > > ### Author Response · Authors · 2024-08-13
> > >
> > > Thanks for the reviewer's efforts and valuable comments.
> > >
> > > ### **Q: Missing orbital view videos**
> > >
> > > **A**: We originally built the anonymous project page following a prior work, i.e. Consistent4D (ICLR2024), that shows the reference view for the demonstration of reconsturction quality and a novel view for that of spatial consistency. And we admit that it would be better to present more novel views in a nice page arrangement. We will improve it.
> > >
> > > As a supplement, we have also provided the video of our composite scene demo in the same project page (i.e. at bottom). The demo video contains surrounding views of the generated dynamic objects, demenstrating their high spatial consistency, as also agreed by reviewer *2HE6*.
> > >
> > > ### **Q: Similar with 4DGS (CVPR2024) on Gaussian attributes deformation**
> > >
> > > **A**: We carefully checked the implementation of 4DGS (CVPR2024). It takes a different approach as ours. In particular, they use a deformation network to directly predict the delta update of each Gaussian's attributes (i.e. position, rotation, scaling etc.). In contrast, we predict the deformation gradients (i.e. transformation matrix) of the control nodes in the deformation graph via an MLP network, the Gaussian attributes are then computed from the predicted node deformation via skinning algorithm in an analytical manner.
> > >
> > > ### **Q: Utilizing of existing 3D representation technology**
> > >
> > > **A**: The authors humbly think the main contributions of the work can be summarized to: 1) the analysis and findings of which prevent existing methods from delivering better performance, as discussed in the reply for "Q4: Insights of this work"; 2) based on these analysis and insights, we for the first time demonstrate the possibility and superior advantages of extending Gaussian-mesh representation for 4D tasks, in comparison to prior methods.
> > >
> > > Although Gaussian-mesh hybrid representation is an existing technology for static scene, no one has really tried to extend it for 4D representation. There is a fact should not be ignored that almost all 4D representations are developed from existing 3D representations. The authors humbly think *a new use of an old method can be novel if nobody ever thought to use it this way for a new problem, and it really can deliver superior performance*.

---

> > > > ### Comment · Reviewer_ow51 · 2024-08-13
> > > >
> > > > Thanks you for the clear answers and I am happy to increase to a positive rating. Based on the novelty and impact of the technology, I can only give it a borderline accept.

---

> > > > > ### Author Response · Authors · 2024-08-13
> > > > >
> > > > > Thank you for your response and support! We sincerely appreciate your efforts and time in reviewing and providing incisive comments on our paper.

---

### Official Review · Reviewer_7V8w · 2024-07-11

**Soundness:** 3
**Presentation:** 3
**Contribution:** 3
**Rating:** 5
**Confidence:** 2

**Summary:**

This paper proposes DreamMesh4D that combines mesh representation with sparse-controlled deformation technique to generate high-quality 4D objects from a monocular video. The authors bind Gaussian splats to the surface of the triangular mesh for differentiable optimization of both the texture and mesh vertices. The method begins with a coarse mesh from single image based 3D generation. Sparse points are then uniformly sampled on surface of the mesh to build a deformation graph, which drives the motion of the 3D object. For each step, transformations of sparse control points are predicted using a deformation network. The mesh vertices and the bound Gaussians are deformed via a geometric skinning algorithm that combines LBS and DQS. Reference view photometric loss, score distillation loss, and regularization losses are used in a two-stage learning. Experiments are performed on the consistent4D dataset.

**Strengths:**

- The methods address a challenging problem to generate dynamic objects from a monocular video.
- A new method is proposed that generates dynamic meshes through a static-to-dynamic optimization process. By employing a Gaussian-mesh hybrid representation, the authors simultaneously refine both the geometry and texture of the object. This approach allows the static object to serve as an excellent starting point for dynamic learning. During the dynamic stage, a deformation graph is used on the object’s surface using geodesic distance.
- Experiments show the superior performance of this method in generating high-fidelity 4D objects.
- The method use dynamic mesh with good compatibility with modern geometric pipelines in the 3D gaming and film industries.

**Weaknesses:**

- One of the main idea of this paper is to use a Gaussian-mesh hybrid representation, which was originally proposed in the work of SuGaR. The technical difference that is designed to address the special setting of this paper’s task is not clearly explained.
- In figure 1, the input is a composited scene video. But in the rest of the paper, the inputs are well segmented object video, which makes the setting of the addressed task confusing.
- The method combines some existing methods such as SuGaR, deformation graph, LBS and DQS based skinning. It’s not clear to identify the author’s innovative contribution.

**Questions:**

1. One of the main idea of this paper is to use a Gaussian-mesh hybrid representation, what is the difference between the proposed method and SuGaR?
2. In figure 1, the input is a composited scene video. Is a segmentation method needed to extract each object region to use as input to the method? There seems no mention on this in the rest of the paper. How is the mutual-occlusion between the objects solved by the proposed method for the scenario in figure 1?
3. What is the computational cost of the proposed method?

**Limitations:**

Not applicable.

---

> ### Author Rebuttal · Authors · 2024-08-06
>
> The authors are grateful for the thoughtful feedback and valuable questions. Thanks for the support for this work! Below we will address the questions and concerns separately.
>
> ### **Q1: The difference between the proposed method and SuGaR.**
> **A**: While SuGaR proposes a pipeline reconstructing static surface via a Gaussian-mesh hybrid representation from pre-trained 3D Gaussians, our work mainly exploit the proposed hybrid representation as our base representation, and focus on extending it for video-to-4D generation task, similarly as most prior works which rely on NeRF or 3D-GS as the base representation.
>
> Although we utilize the static 3D representation proposed by SuGaR, trivially applying it to 4D tasks brings challenges. For example: 1) how to effectively preserve the Gaussian appearance while the asset undergoes deformation (i.e. surface Gaussian attributes update during deformation); 2) how to efficiently and effectively deform the vertices of the mesh-Gaussian representation (i.e. deformation graph and hybrid skinning).
>
> ### **Q2: Concerns about Figure 1.**
> **A**: The input of our method is a monocular video of the target object under a fixed view, which is consistent with the setup used in previous works. As for the composite scene video in Figure 1, it is a demo created by ourselves in Blender using our generated assets. It is to demonstrate that the generated 4D assets by our method can be directly integrated into downstream graphic engines, which further prove the advantage of our work. We will make this more clear in the final version.
>
> ### **Q3: Innovative contribution of this work.**
> **A**: The innovative contribution of our work is that we for the first time demonstrate the possibility and superior advantages of extending Gaussian-mesh representation for 4D tasks, in comparison to prior works, which mainly exploit NeRF or 3D-GS as the underlying representation. As R3 also points out that it is informative to the community that SuGaR works well in a more complex 4D pipeline. Although SuGaR, deformation graph, LBS and DQS are existing techniques, trivially integrating them would not lead to optimal performance. We therefore 1) proposed to change the Gaussian attributes during deformation; 2) proposed a novel skinning algorithm suitable for such optimization-based scenarios with insufficient constrains. The experimental results demonstrate the effectiveness of our design and our method achieves superior performance in comparison to prior baseline methods.
>
> ### **Q4: Computation cost of the proposed method.**
> **A**: The time and memory complexities are 0.8h and 8G for training on a single NVIDIA RTX 6000Ada graphic card, which is also superior compared to prior methods and further demonstrates the advantage of our formulation. The detailed metrics in comparison to other methods can be found in the global rebuttal.

---

> > ### Comment · Reviewer_7V8w · 2024-08-13
> >
> > I appreciate the authors for providing the detailed explanations. For my question 2 for Figure 1, as I guessed, it is a demo created by the authors using their generated assets. I think this should be clearly explained in final version of the paper, because as the current Figure 1, it is easy to misunderstand that the composited scene demo is the input video. For my concerns about the innovation of the paper, the authors explained what are new compared to a straight-forward combination of SuGaR, deformation graph, LBS and DQS. The clarifications are reasonable and are meaningful to work with a pipeline to yield dynamic mesh with good compatibility with modern graphics. In regards of the above, I'd tend to keep my previous positive rating.

---

> > > ### Author Response · Authors · 2024-08-13
> > >
> > > We are pleased to hear that our rebuttal addressed your concerns well, and we are grateful for your efforts and time in reviewing. If you have any further concerns, questions, or suggestions, please do not hesitate to let us know.

---

### Author Rebuttal · Authors · 2024-08-06

We sincerely thank all reviewers for their valuable feedback and agree that:
* we address a challenging problem to generate dynamic objects from a monocular video;
* the method to generate dynamic meshes through a static-to-dynamic optimization process is new, and the losses terms make sense and are intuitive; It was also informative that the SuGaR SDS would work well in a complex pipeline of 4D task;
* the usage of mesh and skinning has good compatibility with modern computer graphic pipelines in the 3D gaming and film industries;
* the experimental results show the superior performance of our method in generating high-fidelity 4D objects compared to previous methods;
* the paper is clearly written, very informative, well contextualized, in good structure and has a good track of the prior methods.

In the following, we will first report both the time and memory complexities of our method against prior works requested by all reviewers, and then address individual reviewer's concerns respectively. All the responses will be incorporated into the final version.

## Comparisons on time and memory complexities.

||Consistent4D|DreamGaussian4D|4DGen|STAG4D|Ours|
|:---:|:---:|:---:|:---:|:---:|:---:|
|Training Time|2.0h|0.6h|3.0h|1.6h|0.8h|
|Memory|28GB|20GB|15GB|7GB|8GB|

The above table shows the computation cost of our method compared to prior works. Our method is efficient in terms of both training time and memory consumption. In particular, our method requires 0.8h and 8G memory during training, which is more efficient than Consistent4D and 4DGen. Although DreamGaussian4D requires 0.6h to train the model, they require much more memory (i.e. 20G). STAG4D requires similar level of memory consumption, however they requires two times more training time than ours. Since Consistent4D cannot run on GPUs with 24GB of memory, we train all methods on a single NVIDIA RTX 6000Ada graphic card with 48GB of memory for fair complexity comparisons. The results further demonstrate the advantage of our method compared to previous works.

---

### Decision · Program_Chairs · 2024-09-25

**Decision:**

Accept (poster)

**Comment:**

This work introduces a novel framework for transforming static images into 4D dynamic meshes, DreamMesh4D. It refines object geometry and texture using a Gaussian-mesh hybrid and a geodesic-based deformation graph. A new skinning algorithm combines the best of LBS and DQS for enhanced deformation. The method excels in creating high-quality 4D objects with improved rendering and consistency.

The work addresses a challenging problem to generate dynamic objects from a monocular video; the method to generate dynamic meshes through a static-to-dynamic optimization process is novel, and the losses terms make sense and are intuitive; It was also informative that the SuGaR SDS would work well in a complex pipeline of 4D task; the usage of mesh and skinning has good compatibility with modern computer graphic pipelines in the 3D gaming and film industries; the experimental results show the superior performance of the proposed method in generating high-fidelity 4D objects compared to previous methods;
the paper is clearly written, very informative, well contextualized, in good structure and has a good track of the prior methods. The rebuttal addresses all the reviewers' concerns.

Overall, the Gaussian-mesh representation for 4D tasks is inspiring and will lead further developments along this direction.